*Report*

# Small-molecule inhibition of aging-associated chromosomal instability delays cellular senescence

Monika Barroso-Vilares[1,2,3], Joana C Macedo[1,2], Marta Reis[1,2], Jessica D Warren[4], Duane Compton[4] & Elsa Logarinho[1,2,*] (ID)

## Abstract

Chromosomal instability (CIN) refers to the rate at which cells are unable to properly segregate whole chromosomes, leading to aneuploidy. Besides its prevalence in cancer cells and postulated implications in promoting tumorigenesis, studies in aneuploidy-prone mouse models uncovered an unanticipated link between CIN and aging. Using young to old-aged human dermal fibroblasts, we observed a dysfunction of the mitotic machinery arising with age that mildly perturbs chromosome segregation fidelity and contributes to the generation of fully senescent cells. Here, we investigated mitotic mechanisms that contribute to age-associated CIN. We found that elderly cells have an increased number of stable kinetochore–microtubule (k-MT) attachments and decreased efficiency in the correction of improper k-MT interactions. Chromosome mis-segregation rates in old-aged cells decreased upon both genetic and small-molecule enhancement of MT-depolymerizing kinesin-13 activity. Notably, restored chromosome segregation accuracy inhibited the phenotypes of cellular senescence. Therefore, we provide mechanistic insight into age-associated CIN and disclose a strategy for the use of a small-molecule to inhibit age-associated CIN and to delay the cellular hallmarks of aging.

**Keywords** aging; chromosomal instability; kinesin-13; kinetochore–microtubule attachments; senescence

**Subject Categories** Cell Cycle; Molecular Biology of Disease

See also: **L Tovini & SE McClelland** (May 2020)

## Introduction

Aging is characterized by the progressive disruption of key biological processes and correlates with the extensive accumulation of macromolecular damage over time. As a consequence, tissue and organ homeostasis is perturbed, which contributes to an overall deterioration of physiological functions. Potential drivers of the aging process have been identified and categorized into hallmarks. Molecular hallmarks comprise DNA damage, telomere attrition, epigenetic remodeling, loss of proteostasis, and mitochondrial dysfunction [1]. Cellular and organismal features of aging include cellular senescence, deregulated nutrient sensing, and stem cell exhaustion [1]. In recent years, several rejuvenation strategies emerged that target these hallmarks [2]. Amongst them, metabolic manipulations and senescent cell ablation (or senolysis) have become popular. Senescent cells, which undergo a permanent cell cycle arrest in response to stressors and exhibit stereotyped phenotypic changes, have been shown to contribute to aging [3,4]. Their targeted clearance was evidenced to attenuate or even prevent age-associated conditions [5–10] and to improve lifespan of naturally aged wild-type mice [11]. Senolysis also extended healthspan in progeroid mice that experience chromosomal instability (CIN) as a result of mitotic checkpoint signaling defects [12]. This suggested a possible link between CIN and aging through the accrual of senescent cells. More recently, this correlation was further supported by studies, showing that loss of chromosome segregation fidelity in otherwise karyotypically stable human cells prompts a CIN-driven senescence signature accompanied by the senescence-associated secretory phenotype (or SASP) [13,14]. Thus, aneuploidy, a state of abnormal chromosome number for long reported to occur with age [15–20], may significantly contribute to the aging process.

Maintenance of chromosomal stability is ensured through the tightly controlled and timely organization of microtubules (MTs) into a bipolar mitotic spindle and microtubule attachment to the complex proteinaceous structures (kinetochores) at the centromeres of all chromosomes prior to their segregation toward opposite poles. Defects in the spindle assembly checkpoint (SAC) that prevents anaphase onset in the presence of unattached kinetochores, as well as the premature separation of sister chromatids due to cohesion defects, will give rise to aneuploid daughter cells [21]. In addition, a major mechanism generating aneuploidy is the persistence of erroneous merotelic kinetochore–microtubule (k-MT) attachments, in which a single kinetochore bound to MTs from opposite poles is left uncorrected, generating an anaphase lagging chromosome and micronuclei (MN) in telophase [22]. As increases in aneuploidy

1 i3S - Instituto de Investigação e Inovação em Saúde, Universidade do Porto, Porto, Portugal
2 Aging and Aneuploidy Group, IBMC—Instituto de Biologia Molecular e Celular, Universidade do Porto, Porto, Portugal
3 Programa doutoral em Biologia Molecular e Celular, Instituto de Ciências Biomédicas Abel Salazar, Universidade do Porto, Porto, Portugal
4 Department of Biochemistry and Cell Biology, Geisel School of Medicine at Dartmouth, Hanover, NH, USA
  *Corresponding author. Tel: +351 220 408 800; E-mail: elsa.logarinho@ibmc.up.pt

have been observed with aging, there is the possibility that the mechanisms required to maintain chromosomal stability might deteriorate with age [23]. Although work on cells and mice with CIN has pointed to a link between chromosomal abnormalities and aging, the mitotic behavior of naturally aged cells was only recently characterized. Analysis of primary human dermal fibroblasts derived from neonatal to octogenarian individuals revealed a progressive loss of proliferative capacity and mitotic dysfunction with age. As a result of the global mitotic gene shutdown caused by the repression of the transcription factor Forkhead box M1 (FoxM1), elderly cells experience chromosome segregation defects that were found to ultimately trigger a full senescence phenotype [24]. Altogether, this raises the intriguing possibility that loss of mitotic fidelity with aging underlies mild CIN, thus favoring the accrual of aneuploid senescent cells and their paracrine effect on the surrounding microenvironment and neighboring cells. Uncovering the yet unknown mechanism(s) by which aging triggers chromosome segregation defects, and resulting aneuploidy, is paramount in light of all recent findings connecting CIN, senescence, and aging.

Here, we show that, in agreement with the previously found mitotic dysfunction, human dermal fibroblasts derived from elderly individuals have lower levels of proteins required for the establishment of proper kinetochore–microtubule (k-MT) attachments, including MT-destabilizing kinesins involved in the correction of merotelic k-MT interactions. As a result of compromised error correction, improper k-MT attachments persist into anaphase giving rise to aneuploid daughter cells. Notably, genetic and pharmacological rescue of MT-destabilizing kinesin-13 activity re-established chromosome segregation accuracy in elderly cells, concomitantly with a reduction in cellular senescence. Consequently, strategic destabilization of k-MT attachments may be a potential strategy to counteract age-associated senescence and thereby act to improve healthspan.

## Results and Discussion

### Defective error correction of improper kinetochore–microtubule (k-MT) attachments in elderly mitotic cells

Minor changes in MT dynamics due to perturbations in the balance of MT stabilizers and destabilizers can profoundly impact the fidelity of chromosome segregation [25]. While a subtle reduction in k-MT stability will delay or prevent the onset of anaphase, small increases in k-MT stability will lead to the persistence of erroneous k-MT attachments (e.g., merotelic) into anaphase. Thus, we tested whether changes in k-MT attachment stability underlie the mild CIN observed with age. We started by comparing kinetochore fibers (k-fiber) in dermal fibroblasts (HDFs) derived from young and elderly healthy Caucasian males (see methods section and [24]). Calcium-induced depolymerization of non-kinetochore microtubules [26] revealed that elderly cells have increased k-fiber intensity levels at the metaphase stage when compared to neonatal cells (Fig 1A and B). Intra- and inter-kinetochore distances of aligned chromosomes in elderly metaphase cells were also increased [27] (Fig 1C and D). However, the stability of k-MT attachments, as measured by the rate of dissipation of fluorescence after photoconversion of Tubulin [28], showed no detectable difference between cells derived from young and elderly individuals (Fig EV1A–C). Taken together, these data

suggest that elderly cells have an increased number of k-MT attachments in metaphase.

If MT occupancy at kinetochores is higher in elderly cells, then an increased number of erroneous k-MT attachments could be expected. To test this prediction, we assessed the efficiency of error correction using the reversible inhibition of kinesin-5 with S-trityl-L-cysteine (STLC) to induce transient monopolar spindles and potentiate the formation of erroneous attachments [29]. Live-cell imaging of cells expressing H2B-GFP/α-Tubulin-mCherry demonstrated that elderly cells are two times more likely to exhibit lagging chromosomes following STLC washout than their young counterparts (11.8% versus 6.3%) (Fig 1E and F). Fluorescence *in situ* hybridization (FISH) analysis for 3 chromosome pairs showed that chromosome mis-segregation is higher in elderly dividing cells (2.22% versus 0.63%) (Fig 1G and H), further confirming the live-cell imaging data. We also measured the frequency of errors upon partial inhibition of Aurora B or Mps1 kinase activities using nanomolar concentration of the small-molecules ZM447439 [30] and AZ3146, respectively. As expected, the frequency of segregation errors significantly increased upon these drug treatments, but a ~2-fold difference was still observed between elderly and neonatal cells (Fig 1E and H; Appendix Fig S1).

Altogether, our data show that aged cells not only generate erroneous k-MT interactions at higher frequency, but also correct them less efficiently. Indeed, gene expression and protein levels of main regulators involved in the establishment of proper k-MT attachments, including the MT-depolymerizing kinesin MCAK, are decreased in elderly cells (Figs 1I and J, and EV2A–J).

### Overexpression of microtubule destabilizing kinesin-13 proteins restores chromosome segregation fidelity

Efficient k-MT error correction relies on the release of microtubules from kinetochores, and two MT-depolymerizing kinesin-13 proteins, Kif2b and MCAK, have been shown to promote k-MT detachment to effect error correction at distinct phases of mitosis [31]. To directly test whether stimulating this machinery affects mild CIN observed with aging, we overexpressed GFP-tagged versions of MCAK (GFP-MCAK) and Kif2b (GFP-Kif2b) in neonatal and elderly cell cultures (Fig 2A, Appendix Fig S2). We found that overexpression of each kinesin-13 rescues the mitotic delay from nuclear envelope breakdown to anaphase onset that we previously reported for advancing age [24] (Fig 2B), indicating that increased kinesin-13 function facilitates mitotic progression. Overexpression of kinesin-13 proteins also reduces the intensity levels of calcium-stable k-fibers in elderly cells (Fig 2C and D). To determine whether increased levels of MCAK or Kif2b ultimately impact chromosome segregation fidelity, we combined a cytokinesis-block assay with FISH staining for 3 chromosome pairs to measure the rates of chromosome mis-segregation in both neonatal and elderly cultures (Fig 2E). Overexpression of MCAK and Kif2b was sufficient to reduce the percentage of elderly binucleated (BN) cells with chromosome mis-segregation, while having no effect on neonatal cells (Fig 2F). Similarly, the frequency of cells with micronuclei (MN; a common fate of anaphase lagging chromosomes) decreased in elderly cells overexpressing the kinesin-13 proteins when compared to control elderly cells (Fig 2G). Taken together, these results indicate that restoring kinesin-13 protein levels suffices to improve error correction and

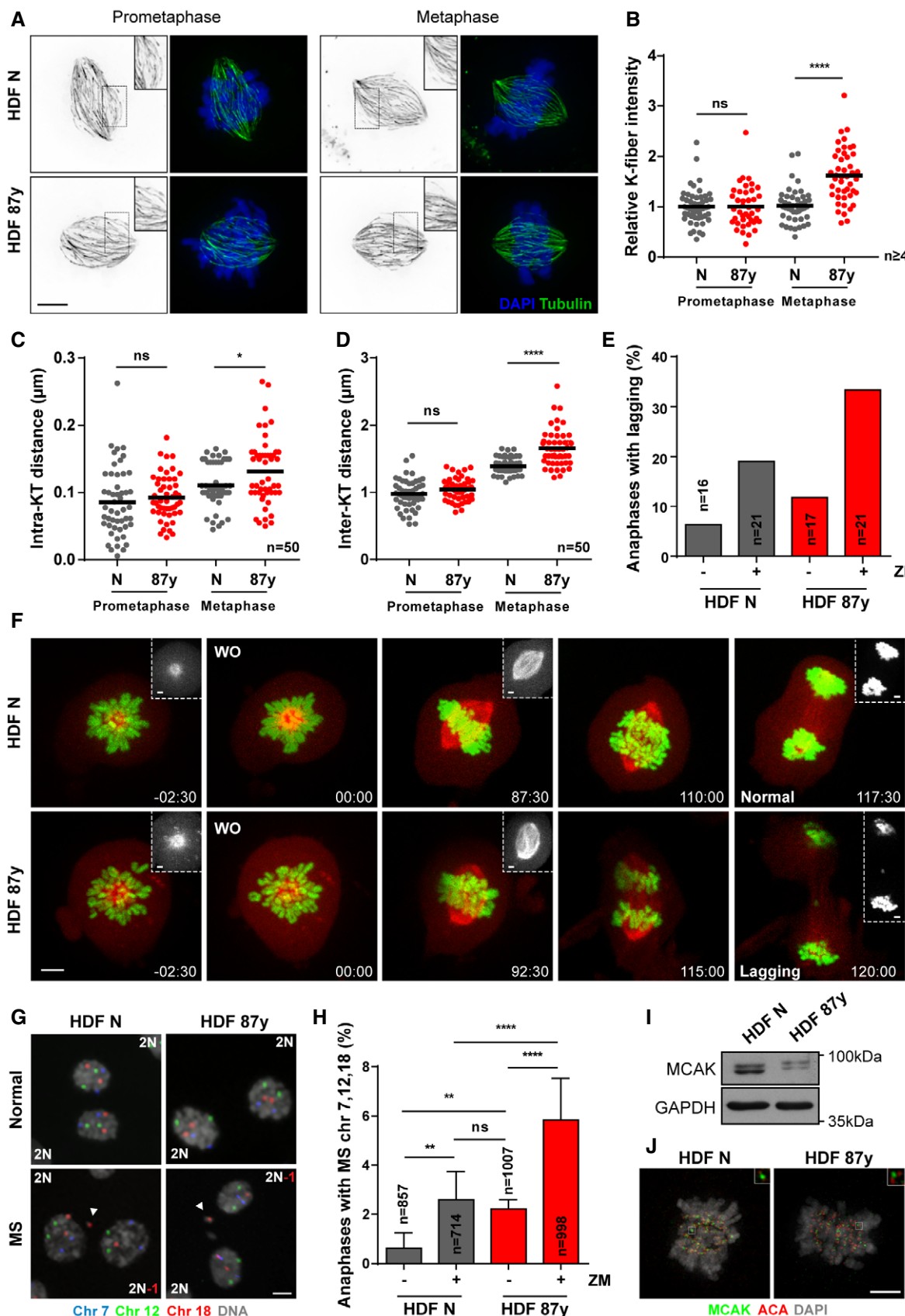

Figure 1.

**Figure 1.   Impaired k-MT error correction with advancing age.**

A, B  Representative images (A) and quantification (B) of calcium-stable k-fiber intensity levels by immunofluorescence analysis of $n \geq 41$ tubulin-stained mitotic cells of neonatal (N) and elderly (87 years) human dermal fibroblasts (HDF) at prometaphase and metaphase stages. Intensity levels were normalized to neonatal samples. Scale bar, 5 μm.

C     Intra-kinetochore distance (between Hec1 and ACA immunostainings of a kinetochore) in $n = 50$ kinetochore pairs scored from 10 elderly versus neonatal cells in prometaphase and metaphase.

D     Inter-kinetochore distance (between Hec1 staining of sister kinetochores) in $n = 50$ kinetochore pairs scored from 10 elderly versus neonatal cells in prometaphase and metaphase.

E, F  Live-cell imaging of neonatal (N) and elderly (87 years) fibroblasts expressing H2B–GFP/α-Tubulin-mCherry treated with kinesin-5 inhibitor (STLC) to induce monopolar spindles, followed by washout (WO) into medium with DMSO (−) or ZM447439 (+; 500 nM). (E) Quantification of anaphases with lagging chromosomes in $n =$ cells scored and (F) representative movie frame series of a young and an elderly dividing cell that underwent correct (Normal) and incorrect (Lagging) chromosome segregation, respectively. Time, min:sec. Scale bars: 5 μm (images) or 2 μm (insets).

G     Representative images of anaphases without (top) and with (bottom) mis-segregation (MS), FISH-stained for three chromosome pairs (7, 12, and 18). Arrowheads indicate micronuclei containing centromeric signal for chromosome 18. Scale bar, 10 μm.

H     Percentage of anaphases with MS in neonatal (N) versus elderly (87 years) $n =$ cells scored by FISH analysis.

I     Western blot analysis of total MCAK protein levels in neonatal (HDF N) and elderly (HDF 87 years) fibroblasts. GAPDH is shown as loading control.

J     Immunofluorescence analysis of MCAK levels in neonatal (HDF N) and elderly (HDF 87 years) mitotic cells. Scale bar, 5 μm.

Data information: All values shown are mean ± SD of at least two independent experiments. ns $P > 0.05$, *$P < 0.05$, **$P < 0.01$, ****$P < 0.0001$ by two-tailed (B–D) Mann–Whitney test and (H) chi-square test.
Source data are available online for this figure.

mitotic fidelity in elderly cells, confirming that defective error correction of improper k-MT attachments is significantly contributing to the mild CIN observed with aging.

**Strategic destabilization of k-MTs delays senescence**

Lagging chromosomes and the ensuing MN were recently recognized as a source of pro-inflammatory signaling when loss of compartmentalization exposes MN chromatin to cytosolic cyclic GMP-AMP synthase (cGAS) [32–34], a local DNA sensor to initiate innate immune response against foreign DNA [35,36], and more recently also reported as an essential player for cellular senescence [37–39]. In a previous study, we found chromosome mis-segregation to be a key trigger for the development of full senescence phenotypes in elderly cells [24], which consistently with an inflammatory gene expression profile also exhibit increased secretion of cytokines (Fig EV3A). Thus, we tested whether counteracting age-associated mild CIN could delay the development of full senescence in elderly cells.

Analysis of cGAS gene expression revealed no significant changes between young and elderly fibroblast cultures (Fig EV3B), but we found that the percentage of cells with cGAS-positive MN is significantly increased in elderly cultures (Fig EV3C and D). cGAS-

positive MN were typically Rb-negative, thus suggesting nuclear envelope was disrupted (Fig EV3C and D) [33]. The percentage of elderly cells with cGAS-negative/Rb-positive MN was also increased, likely representing a fraction of MN that still did not undergo nuclear envelope disruption. In agreement with the role of this cytosolic DNA sensor in pro-inflammation and senescence development, depletion of cGAS (Fig EV3E) was able to decrease the percentage of cells with cGAS-positive MN without changing the total percentage of MN (Fig EV3F), and to reduce the senescence-associated phenotypes in elderly cells (Fig EV3G and H). These results indicate that cGAS engagement induced by mild CIN may play an important role in the development of cellular senescence with aging. To further test this idea, we analyzed the implications of kinesin-13 overexpression on MN cGAS positivity and senescence. Increased levels of MCAK and Kif2b led to decreased frequency of MN and equivalently fewer staining positive for cGAS (Fig EV3I). The percentage of cells exhibiting senescence biomarkers was reduced upon improved error correction efficiency (Fig 3A–D). Moreover, targeted transcriptomic analysis (see methods section; Fig 3E and F; Dataset EV1) revealed marked improvements in senescence-associated gene expression profile. From a custom list of senescence and SASP-related genes (Fig 3G; Dataset EV2), differential expression between young and elderly cells according to the

**Figure 2.   Overexpression of MT depolymerases, MCAK and Kif2b, restores chromosome segregation fidelity in elderly cells.**

A     GFP protein levels in FACS-sorted neonatal (N) and elderly (87 years) fibroblasts transduced with empty (control, −), GFP-MCAK (+), and GFP-Kif2b (+) lentiviral plasmids. Tubulin is shown as loading control.

B     Mitotic duration scored by time-lapse phase-contrast microscopy of $n \geq 50$ cells per condition measured from nuclear envelope breakdown (NEB) to anaphase onset.

C, D  Representative images (C) and quantification (D) of calcium-stable k-fiber intensity levels by immunofluorescence analysis of $n \geq 26$ tubulin-stained metaphase cells transduced as indicated. Levels were normalized to the control neonatal sample. Scale bar, 5 μm.

E     Representative images of cytochalasin D-induced binucleated (BN) cells without (top) and with (bottom) mis-segregation (MS), FISH-stained for three chromosome pairs (7, 12, and 18). Scale bar, 10 μm.

F     Percentage of BN cells with MS in $n =$ cells scored.

G     Percentage of micronuclei in $n =$ cells analyzed.

Data information: All values are mean ± SD of at least two independent experiments. ns $P > 0.05$, *$P < 0.05$, **$P < 0.01$, ****$P < 0.0001$ by two-tailed (B, D) Mann–Whitney and (F, G) chi-square test.
Source data are available online for this figure.

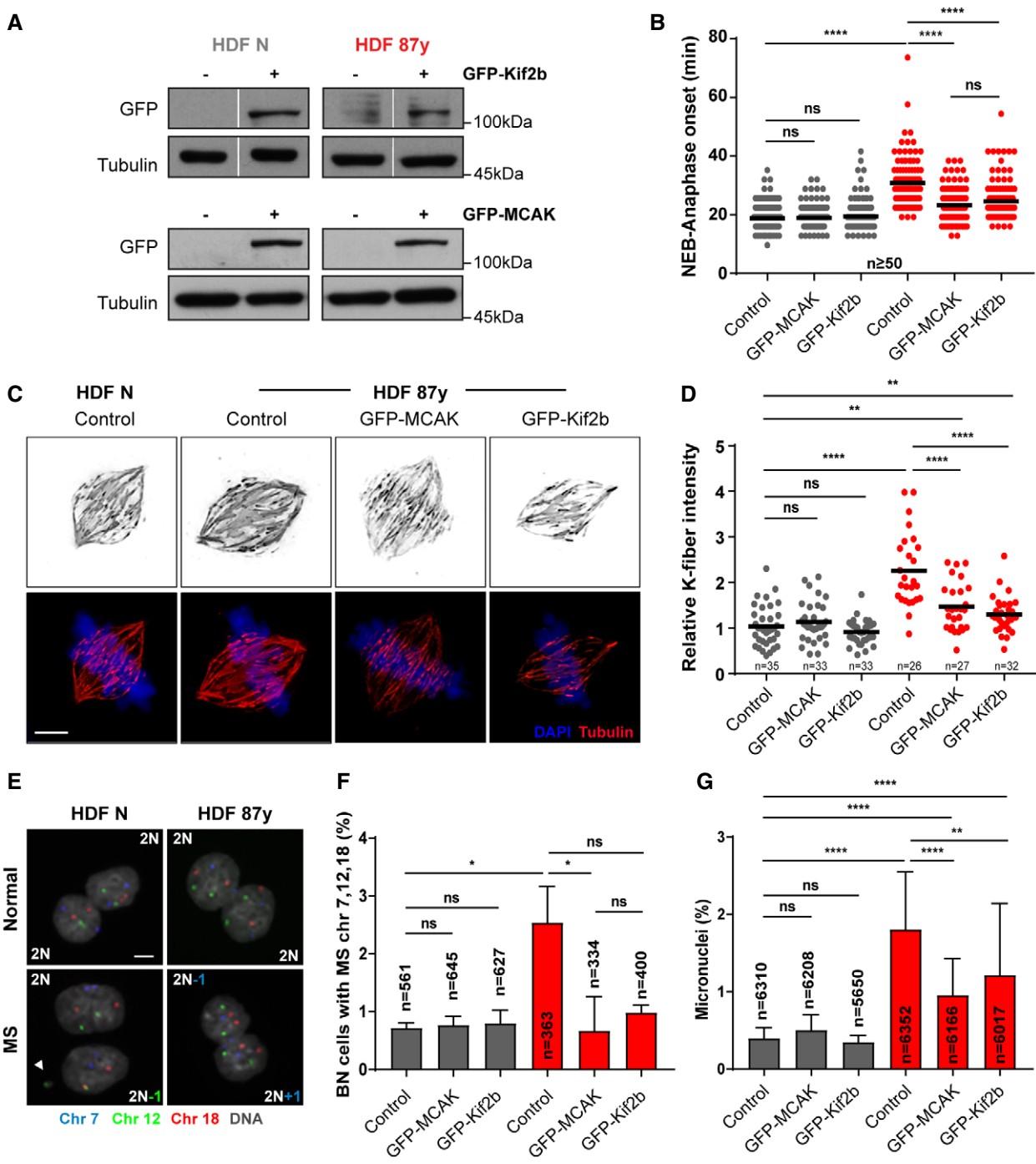

**Figure 2.**

expected was observed for 20 genes, in which 17 were partially recovered following overexpression of kinesin-13 proteins. We additionally extended the analysis to a set of genes that defines a "senescence core signature" common to different cell types and senescence-inducing stimuli [40]. Comparison between octogenarian and neonatal cells revealed 23 genes significantly altered as expected, out of which 17 were modulated upon overexpression of MCAK and Kif2b in elderly cells (Appendix Fig S3, Dataset EV3). Moreover, we interrogated the transcriptome dataset for a custom

list of 56 genes associated with the cGAS/STING/NF-κB pathway (Dataset EV4), and we found that out of the 25 genes differentially expressed in octogenarian versus neonatal cells accordingly to the expected, 23 partially rescued by MCAK and/or Kif2b overexpression in old-aged cells (Fig 3H). Taken together, these data demonstrate that kinesin-13 protein overexpression in aged cells restores chromosome segregation fidelity and reduces the burden of MN-induced pro-inflammatory cGAS signaling. This in turn has a positive impact on the elderly cell population, as significant

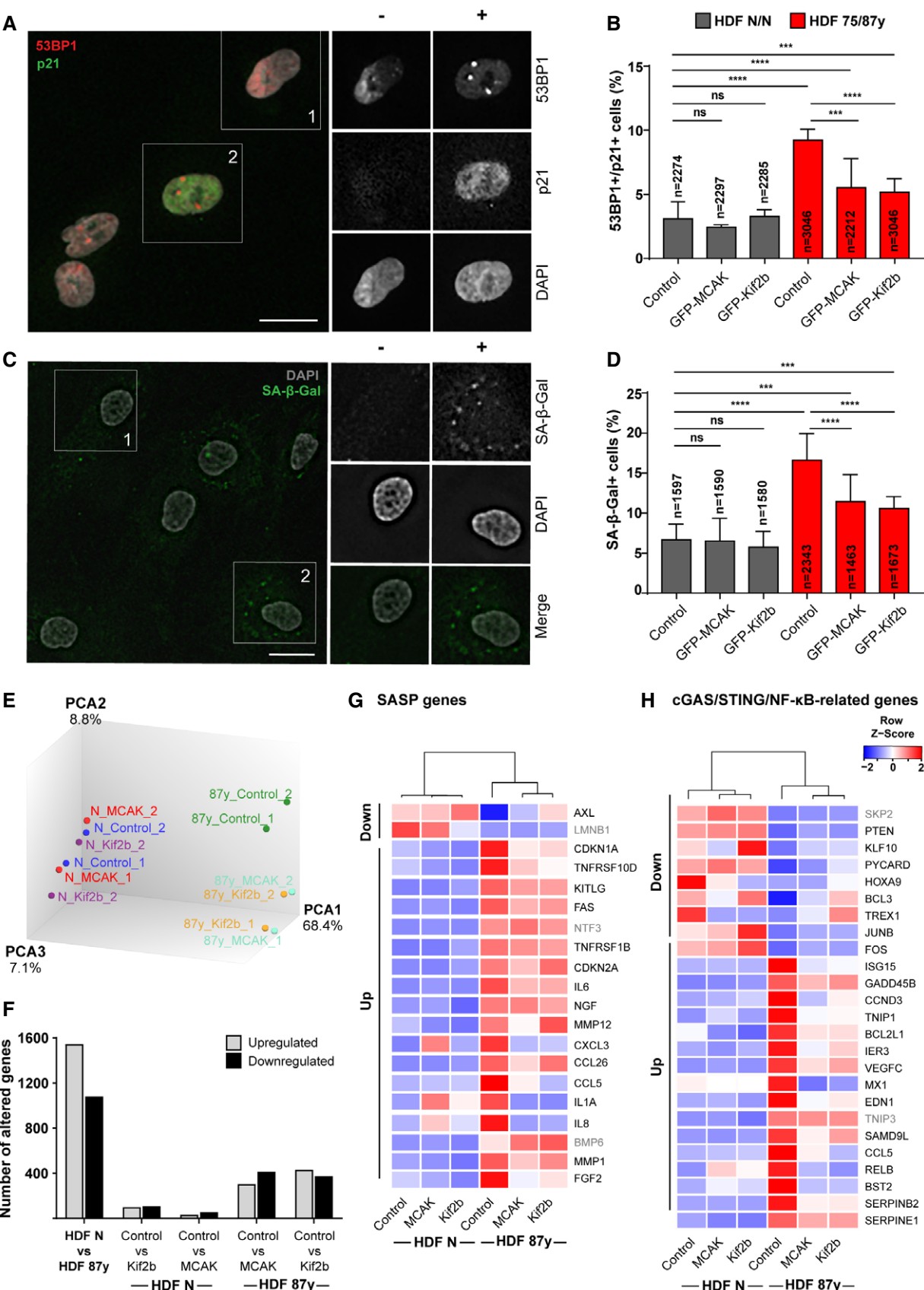

Figure 3.

◀

**Figure 3.  Overexpression of MCAK and Kif2b delays senescence in fibroblast cultures from elderly donors.**

A    Representative images of elderly cells scored as negative (−) or positive (+) for Cdkn1a/p21 (cell cycle inhibitor) and 53BP1 (≥ 1 foci; DNA damage) senescence biomarkers. Scale bar 20 μm.

B    Percentage of $n$ = cells staining double-positive for Cdkn1a/p21 and 53BP1 in neonatal (N/N) and elderly (75/87 years) human dermal fibroblasts (HDF) transduced with empty, GFP-MCAK or GFP-Kif2b lentiviral plasmids.

C    Representative images of elderly cells scored as negative (−) or positive (+) for SA-β-galactosidase (SA-β-gal) activity. Scale bar, 20 μm.

D    Percentage of $n$ = cells positive for SA-β-gal in neonatal and elderly HDFs transduced with empty, GFP-MCAK or GFP-Kif2b lentiviral plasmids.

E    Principle component analysis (PCA) of neonatal and elderly cells transduced with empty-vector (control) or kinesin-13 proteins (MCAK or Kif2b) based on bulk RNA expression data.

F    Total number of genes altered (up- or downregulated, as indicated) for the comparisons shown between control and kinesin-13-overexpressing young (HDF N) and/or aged (HDF 87 years) cells.

G, H   Heatmaps of differentially expressed (G) SASP and senescence genes, and (H) cGAS/STING/NF-κB-related genes of neonatal and 87 years HDFs. Down and Up refer to the expected changes as reported for senescent cells (Datasets EV2 and EV4). Gene symbols highlighted in gray indicate genes that were not modulated by overexpression of kinesin-13 proteins. $Z$-score row color intensities indicate higher (red) to lower (blue) expression.

Data information: Values are mean ± SD of at least two independent experiments. ns $P > 0.05$, ***$P < 0.001$, ****$P < 0.0001$ by two-tailed chi-square test (B, D).

improvements in the senescence-associated transcriptome signature match with delayed emergence of fully senescent cells permanently arrested in the cell cycle.

## Small-molecule inhibition of age-associated chromosome mis-segregation delays senescence

A recent study identified a small-molecule that specifically potentiates the activity of the kinesin-13 protein MCAK (UMK57), to transiently suppress chromosome mis-segregation in CIN$^+$ cancer cells [41]. Since perturbed error correction contributes to mild CIN in aged cells, we reasoned that this agonist should provide a small-molecule approach, as an alternative to genetic overexpression, to suppress CIN. Thus, we followed cells exposed to increasing concentrations of UMK57 under 24-h long-term time-lapse microscopy and found that 1 μM is sufficient to rescue the increased mitotic duration in elderly cells, while having no noticeable effect on the mitotic progression of neonatal cultures (Fig 4A). As 1 μM is a 10× higher dose than the previously reported for CIN$^+$ cancer cells, we fine-tuned the titration to lower concentrations using phenotypic readouts for CIN and senescence (Fig EV4A–D). We found that 1 μM is indeed required to significantly rescue the age-associated phenotypes. Noteworthy, 1 μM UMK57 did not overtly change the MCAK protein levels in neonatal or elderly cells (Fig EV4E). Also, we showed that UMK57 doses > 0.1 μM are unable to suppress, or even increase, CIN in HT-1080 fibrosarcoma cells, which have 60% higher MCAK levels than neonatal fibroblasts (Fig EV4F–I). This supports the safety use of 1 μM UMK57 in the context of untransformed aged cells, which is different from the context of cancer cells, which typically overexpress MCAK. All further experiments were conducted with this optimal dose of UMK57. In agreement with the rescued mitotic delay, calcium-stable k-fiber intensity analysis in UMK57-treated elderly cells revealed that enhanced MCAK activity decreases the number of stable k-MT attachments in metaphase (Fig 4B and C). FISH analysis of 3 chromosome pairs in both interphase cells and cytokinesis-blocked BN cells showed that UMK57 decreases the levels of aneuploidy and the chromosome mis-segregation rate in elderly cell populations (Fig 4D and E). Also, MN levels were scored and found to be partly decreased in elderly cells upon 24-h exposure to the MCAK agonist (Fig 4F). These results indicate that age-associated mild CIN can be rescued using a small-molecule agonist of the kinesin-13 protein

MCAK, thus supporting that reduced error correction contributes to mild CIN with age.

We next tested whether UMK57-induced MCAK activity could delay cellular senescence. We found that 24-h treatment is sufficient to partially rescue the percentage of cells exhibiting the senescence biomarkers 53 bp1+p21 and SA-β-galactosidase activity (Fig 4G and H). Together, these results demonstrate that strategic destabilization of k-MT attachments aids in the correction of improper k-MT attachments, while acting to counteract MN-induced pro-inflammation and cellular senescence with aging.

Efforts to suppress CIN in cancer cell lines using UMK57 unveiled a rapidly arising adaptive resistance mechanism, whereby reversible rewiring of mitotic signaling networks abolishes the CIN-inhibiting effect of the compound [41]. To explore whether such adaptive resistance mechanisms arise in our cellular model of aging, we exposed cells to UMK57 for longer periods. After 96-h treatment, the mitotic delay of elderly cells was rescued (Fig 5A), indicating that the beneficial effect of UMK57 persists over long-term exposure. In agreement, decreased k-fiber intensity levels in metaphase were still observed after 96 h (Fig 5B). FISH analyses on interphase and BN cells demonstrated that aneuploidy (Fig 5C), chromosome mis-segregation (Fig 5D), and micronucleation (Fig EV5A) were also inhibited after 96 h. Furthermore, we found that the decrease in MN, including cGAS$^+$ MN (Fig EV5B), correlates with a repression of cellular senescence, demonstrated by the partial rescue in senescence markers (Fig 5E and F) and mild changes in gene expression of senescence-associated genes, including genes of the "senescence core signature" (Fig EV5C). Similar observations were taken for 4 weeks of exposure to UMK57 (Fig EV5D), for which decreased levels of aneuploidy and senescence markers were also observed (Fig EV5E–G).

Adaptive resistance arising in CIN$^+$ cancer cell lines was shown to be Aurora B-dependent, the activity of which dropped significantly with longer exposures and correlated with increases in k-MT attachment stability specifically in prometaphase [41]. In contrast to CIN$^+$ cancer cells, we found that in elderly cells exposed to UMK57 for 96 h, k-fiber intensity levels in prometaphase were unaltered (Fig EV5H), even though analysis of active Aurora B kinase (T232 phosphorylation) at centromeres revealed oscillating levels, a slight decrease in neonatal cells at 96 h and an increase in aged cells at 24 h but not at 96 h (Fig EV5I). In metaphase, we found slightly reduced levels of active Aurora B in aged cells at 96 h (Fig 5G and H), albeit

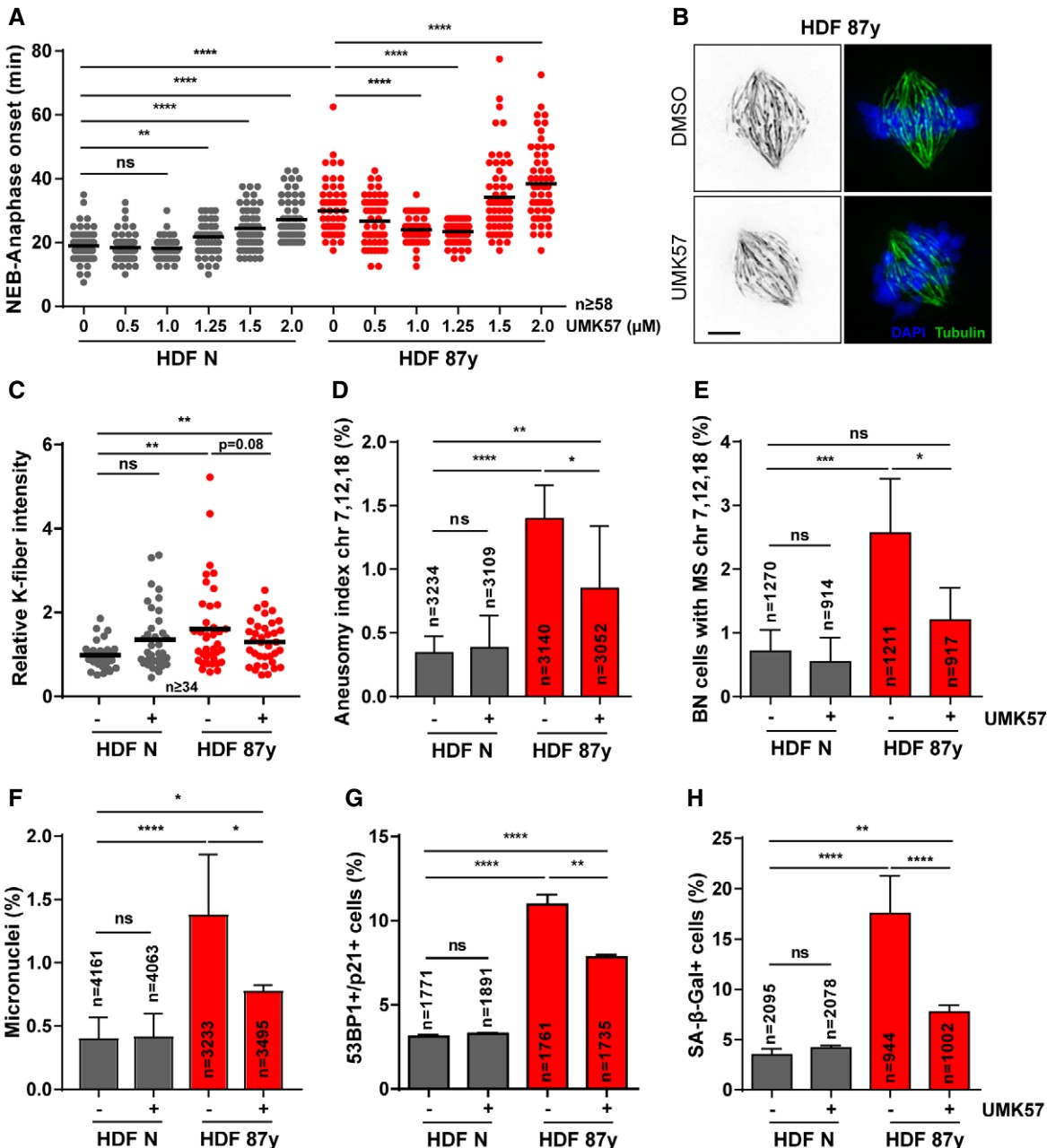

**Figure 4.  Small-molecule agonist of MCAK activity rescues age-associated CIN and delays senescence.**

A   Mitotic duration of neonatal (N) and elderly (87 years) human dermal fibroblasts (HDF) treated for 24 h with different concentrations of UMK57 (MCAK agonist). $n \geq 58$ cells were analyzed per condition. For all subsequent experiments, UMK57 was used at 1 µM for 24 h.

B, C   Representative images (B) and quantification (C) of calcium-stable k-fiber intensity levels in metaphase, scored by immunofluorescence analysis of $n \geq 34$ tubulin-stained mitotic cells of neonatal and elderly samples treated with DMSO (−) and UMK57 (+). Levels were normalized to neonatal DMSO-treated condition. Scale bar, 5 µm.

D   Aneusomy index of chromosomes 7, 12, and 18 measured by interphase FISH analysis of $n =$ cells.

E   Percentage of cytochalasin D-induced binucleated (BN) $n =$ cells with chromosomes 7, 12, and 18 mis-segregation (MS).

F   Percentage of micronuclei in $n =$ cells scored when treated with DMSO or UMK57.

G   Percentage of $n =$ cells staining positive for double immunostaining of Cdkn1a/p21 (cell cycle inhibitor) and 53BP1 (≥ 1 foci; DNA damage) senescence biomarkers.

H   Percentage of $n =$ cells staining positive for SA-β-galactosidase (SA-β-gal) activity.

Data information: All values are mean ± SD of at least two independent experiments. ns $P > 0.05$, *$P < 0.05$, **$P < 0.01$, ***$P < 0.001$, ****$P < 0.0001$ by two-tailed (A, C) Mann–Whitney and (D–H) chi-square tests.

   

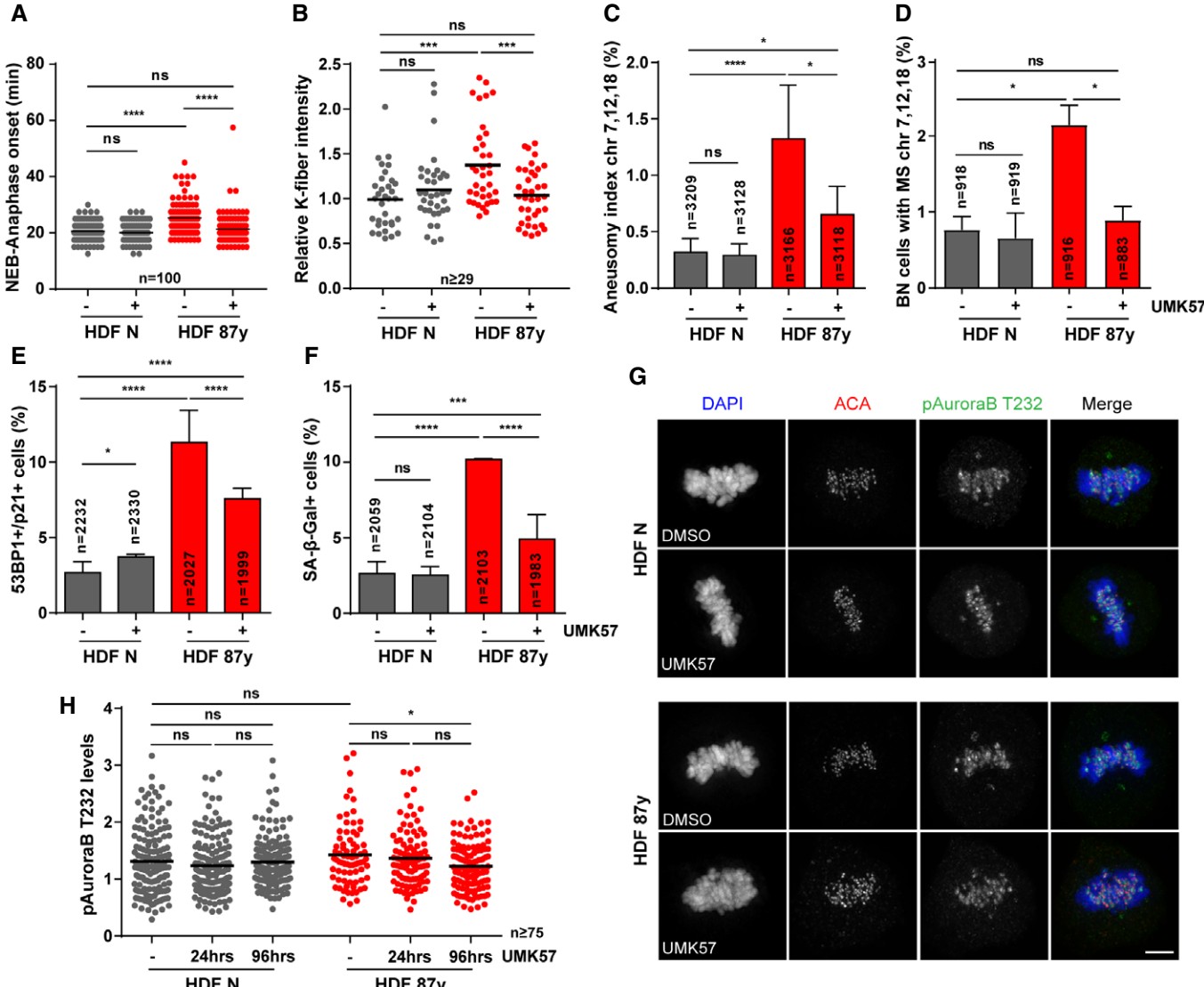

**Figure 5. Long-term treatment with UMK57 does not lead to adaptive resistance in elderly cells.**

A   Mitotic duration of $n = 100$ neonatal (N) and elderly (87 years) human dermal fibroblasts (HDF) treated with DMSO (−) or UMK57 (+) for 96 h.

B   Relative calcium-stable k-fiber intensity levels scored by immunofluorescence analysis of $n \geq 29$ tubulin-stained metaphase cells of neonatal and elderly samples treated with DMSO and UMK57 for 96 h. Levels were normalized to neonatal DMSO-treated condition.

C   Aneusomy index of chromosomes 7, 12, and 18 measured by interphase FISH analysis of neonatal and elderly $n =$ cells treated for 96 h.

D   Cytochalasin D-induced binucleated (BN) $n =$ cells with mis-segregation (MS) of chromosomes 7, 12, and 18 in neonatal and elderly samples treated for 96 h.

E   Percentage of $n =$ cells staining positive for double immunostaining of Cdkn1a/p21 (cell cycle inhibitor) and 53BP1 ($\geq 1$ foci; DNA damage) senescence biomarkers after 96 h of treatment.

F   Percentage of $n =$ cells staining positive for SA-β-gal activity when treated for 96 h with DMSO or UMK57.

G, H   Representative images (G) and quantification (H) of phospho-Aurora B Thr232 (pAuroraB T232) levels at kinetochores/centromeres in $n \geq 75$ neonatal and elderly metaphase cells treated with UMK57 for 24 or 96 h. Intensity levels were normalized to ACA. Scale bar, 5 μm.

Data information: All values are mean ± SD of at least two independent experiments. ns $P > 0.05$, *$P < 0.05$, ***$P < 0.001$, ****$P < 0.0001$ by two-tailed (A, B, H) Mann–Whitney and (C–F) chi-square tests.

not translated into an adaptive response by functional analysis (Fig 5B–D). Taken together, these results indicate that suppression of mild CIN using this small-molecule agonist of MCAK may be a potential strategy to counteract age-associated senescence.

The mechanistic link between chromosomal instability (CIN) and senescence in the context of aging has remained elusive. Based on our previous findings [24], we reasoned that defective k-MT

attachments could contribute to the mild levels of CIN observed in mitotically active cell populations from elderly donors. We found that calcium-stable k-fiber intensity and inter-/intra-kinetochore distance are increased in aged fibroblasts, specifically in metaphase. These data suggest that an increased number of k-MT attachments could be responsible for the increased number of lagging chromosomes and micronuclei observed in elderly cell populations.

The importance of MT occupancy for the proper segregation of merotelically attached kinetochores has been previously demonstrated in non-cancerous human cells [42]. Increased microtubule occupancy at kinetochores may arise as a result of altered microtubule assembly rates [43], or altered kinetochore architecture. Cells from aged individuals were reported to exhibit peri/centromeric satellite heterochromatin decondensation, an early indicator of cellular senescence [44]. The levels of the histone H3 variant centromeric protein A (CENP-A), which epigenetically determines human centromeres, are known to be decreased in senescent cells [24]. Despite the reduction of CENP-A expression, our previous transcriptomic analysis showed that CENP-C, a crucial component for kinetochore assembly, is upregulated in elderly cells [24]. Thus, a possibility is that distension of centromeres combined with increased CENP-C levels may provide a structural platform capable of establishing a higher MT occupancy in aged cells. With more MT attachments per kinetochore in elderly cells, the normal rate of k-MT detachment in metaphase is less likely to be sufficient in correcting all erroneous attachments [45]. In support of this idea, increasing k-MT detachment rate, through the overexpression of kinesin-13 proteins or through treatment with UMK57, rescued k-fiber intensity levels, improved error correction and segregation fidelity in the aged fibroblasts, and ultimately prevented the generation of cGAS-positive micronuclei. Therefore, impaired kinesin-13 activity can be established as a mechanistic link between chromosome mis-segregation and senescence in naturally aged cells, by significantly contributing to cGAS-dependent pro-inflammatory response.

Here, we demonstrated that pharmacological rescue of MCAK activity is a means to delay cellular aging. Importantly, the optimal concentration of UMK57 that improved chromosome segregation fidelity in elderly cells did not impact the mitotic fitness of younger cells. Also, we excluded the possibility of adaptive resistance to prolonged UMK57 treatment, which has previously impaired its clinical application in CIN$^+$ cancer cells [41]. Even though modulation of CIN solely acts on mitotically active aged cell/tissue populations, there are substantial arguments for it to be taken into consideration as an anti-aging strategy. First, different types of proliferative cells support the function of stem and differentiated cell pools via paracrine signaling, by secreting bioactive molecules [10]. Also, emergent rejuvenation strategies such as dietary regimens, cellular reprogramming, and senolysis, primarily target cells with proliferative potential/capacity (adult stem cells, vascular and connective tissue cells) or with loss of proliferative capacity (senescent cells) [2]. Second, modulation of CIN encompasses several advantages of senolysis since it prevents the generation of fully senescent cells and their detrimental paracrine signaling. Notably, it bypasses hurdles that still dim the applicability of senolytic therapies, including the need to determine a critical time window for treatment onset and the off-target effects on beneficial senescence [46]. Finally, reducing the burden of SASP factors via CIN modulation likely has a positive impact on the changes observed in the immune system with age (immunosenescence), which have been ascribed to persistent pro-inflammation in tissues [47]. The here shown small-molecule modulation of kinesin-13 MCAK acts precisely upstream in the order of events, i.e., CIN–micronuclei–cytosolic DNA–cGAS/STING activation–SASP, by reestablishing mitotic competence and diluting out senescent cells [48]. Consequently, *in vivo* studies will be paramount to determine the overall impact of chromosome segregation improvement over time at the organismal level.

# Materials and Methods

### Cell culture

Human dermal fibroblasts (HDFs) retrieved from skin samples of neonatal (No. GM21811, Coriell Institute; No. DFM021711A, Zen Bio) and octogenarian (No. AG07135; AG13993; AG09271; AG10884; all from Coriell Institute) Caucasian males reported as "healthy" were grown in minimal essential medium Eagle–Earle (MEM) supplemented with 15% fetal bovine serum (FBS), 2 mM L-glutamine, and 1× antibiotic–antimycotic (all from Gibco, Thermo Fisher Scientific). Only early passage dividing fibroblasts (up to passage 3–5) with cumulative population doubling level (PDL) < 24 were used. HT-1080 (ATCC®, CCL-121™) cells were cultured in MEM supplemented with 10% FBS, 2 mM L-glutamine, and 1× antibiotic–antimycotic (all from Gibco, Thermo Fisher Scientific).

### Drug treatments

Proteasome inhibitor MG-132 (474790, EMD Millipore) was used at 5 μM for 2 h to arrest cells at the metaphase stage. Cytochalasin D (C8273, Sigma-Aldrich) was used at 1 μM for 24 h to block cytokinesis. Fibroblasts were treated with 2.5 μM STLC (2191, TOCRIS) for 5 h to inhibit kinesin-5 activity and induce monopolar spindles, followed by a washout into fresh medium with 500 nM of Aurora kinase B inhibitor ZM447439 (S1103, Selleckchem) to potentiate chromosome segregation errors. To enrich the Mitotic Index for mitotic cell shake-off, STLC was used at 5 μM during 16 h. To partially inhibit Mps1 kinase activity, 500 nM of AZ3146 (3994, TOCRIS) was used during 4 h. 1 μM of UMK57 (kindly provided by Dr. Benjamin Kwok) was used to enhance kinesin-13 activity during the time periods indicated for each experiment.

### Lentiviral plasmids

To assemble pLVX-Tight-Puro plasmids for lentiviral transduction and expression of GFP-MCAK and mEOS-α-Tubulin, BamHI-NotI-tailed fragments were PCR-amplified from GFP-MCAK (gift from Dr. Linda Wordeman) and mEos2-Tubulin-C-18 (#57432, Addgene), respectively. To generate pLVX-Tight-Puro-GFP-Kif2b, a NotI-MluI-tailed fragment was amplified from GFP-Kif2b (gift from Dr. Linda Wordeman). The PCR products were then ligated into the BamHI and NotI, or NotI and MluI restriction sites of digested pLVX-Tight-Puro vector (Clontech). All primers used for PCR amplifications are listed in Appendix Table S1.

### Lentiviral production and infection

Lentiviruses were produced according to the Lenti-X Tet-ON Advanced Inducible Expression System (Clontech). HEK293T helper cells were transfected with packaging plasmids pMd2.G and psPAX2 using Lipofectamine 2000 (Life Technologies) to generate responsive lentiviruses carrying pLVX-Tight-Puro, pLVX-Tight-Puro-H2B-GFP/α-Tubulin mCherry [24], pLVX-Tight-Puro-GFP-MCAK,

pLVX-Tight-Puro-GFP-Kif2b, or pLVX-Tight-Puro-mEOS-α-Tubulin, as well as transactivator lentiviruses carrying the rtTA expressing construct (pLVX-Tet-On Advanced). Human fibroblasts were then co-infected for 6 h with both the responsive and the transactivator lentiviruses (2:1 ratio) in the presence of 8 µg/ml polybrene (AL-118, Sigma-Aldrich). Co-transduction was induced with 500 ng/ml doxycycline (D9891, Sigma-Aldrich). Transfection efficiencies of all experiments were determined by scoring the number of fluorescent cells, or protein levels by Western blot analysis.

## Fluorescence-activated cell sorting

Subpopulations of GFP-positive cells were FACS sorted to validate lentiviral transduction of pLVX-Tight-Puro-GFP-MCAK, pLVX-Tight-Puro-GFP-Kif2b, and pLVX-Tight-Puro-EOS-α-Tubulin. FACS sorting was performed using a FACSAria™ I Cell Sorter (BD Biosciences), with the laser line of 488 nm. Dead cells and subcellular debris were excluded using gates based on forward scatter area (FSC-A) versus side scatter area, while cell doublets and clumps were excluded through FSC-A versus FSC-width plot. The signal was detected using the APC-A channel and gates designed based on the respective auto-fluorescent control.

## cGAS siRNA knockdown

Cells were plated and after 1 h transfected with siRNA oligonucleotides targeting cGAS (Dharmacon, M-015607-01-0005, siGENOME 115004) at a final concentration of 25 nM. Transfections were performed using Lipofectamine RNAiMAX in Opti-MEM medium (both from Thermo Fisher Scientific) according to the manufacturer's instructions. Transfection medium was replaced with complete medium after 6 h. All experiments were performed 72 h post-transfection, and protein depletion confirmed by Western blot and immunostaining analyses.

## Fluorescence *in situ* hybridization

FISH was used to score aneusomy indexes (interphase FISH; Figs 4D and 5C, and EV5E) and chromosome mis-segregation (MS) rates. MS rates were scored by Cyto-D FISH (Figs 2E and F, and 4E and 5D and Appendix Fig S1), or by FISH on STLC-washed-out fibroblasts (Fig 1G and H). For all experiments, fibroblasts were grown on Superfrost™ Plus microscope slides (Menzel, Thermo Fisher Scientific) placed in quadriPERM dishes (Sarsted). Cells were fixed with ice-cold Carnoy fixative (methanol:glacial acetic acid, 3:1), following an initial 30-min hypotonic shock in 0.03 M sodium citrate solution (Sigma-Aldrich). FISH was performed with Vysis centromeric probes CEP7 Spectrum Aqua, CEP12 Spectrum Green, and CEP18 Spectrum Orange (all from Abbott Laboratories) according to manufacturer's instructions. DNA was counterstained with 0.5 µg/ml 4′,6-diamidino-2-phenylindole (DAPI), and microscope slides were then mounted with coverslips in proper anti-fading medium (90% glycerol, 0.5% N-propyl gallate, 20 mM Tris pH = 8.0).

## SA-β-gal assay

Cells were incubated in culture medium containing 100 nM Bafilomycin A1 (B1793, Sigma-Aldrich) for 90 min to induce lysosomal alkalization. The fluorogenic substrate for β-galactosidase, fluorescein di-β-D-galactopyranoside (33 µM; F2756, Sigma-Aldrich) or DDAO galactoside (10 µM; Setareh Biotech LLC), was subsequently added to the medium for 90 min. Cells were fixed in 4% paraformaldehyde for 15 min, rinsed with PBS, and permeabilized with 0.1% Triton X-100 in PBS for 15 min. 0.5 µg/ml of DAPI (Sigma-Aldrich) was used to counterstain DNA, and coverslips were then mounted on slides.

## Calcium-stable k-fiber analysis

Fibroblasts grown on sterilized glass coverslips coated with 50 µg/ml fibronectin (F1141, Sigma-Aldrich) were incubated in calcium buffer (100 mM PIPES, 1 mM $MgCl_2$, 1 mM $CaCl_2$, 0.5% Triton X-100, pH = 6.8) for 5 min and fixed with 4% paraformaldehyde/0.25% glutaraldehyde in PBS for 15 min, both at 37°C. Next, cells were rinsed first in PBS, then in TBS (50 mM Tris–HCl, pH = 7.4, 150 mM NaCl), and permeabilized in TBS + 0.3% Triton X-100 for 7 min. Blocking was performed with 10% FBS + TBS + 0.05% Tween-20 for 1 h, and cells were then incubated with mouse anti-α-Tubulin (T5168, Sigma-Aldrich) antibody diluted at 1:1,500 in 10% FBS + TBS + 0.05% Tween-20. The secondary antibodies Alexa Fluor-488 and Alexa Fluor-568 (Life Technologies) were used at 1:1,500 in 5% FBS + TBS + 0.05% Tween-20. DNA was counterstained with 0.5 µg/ml DAPI (Sigma-Aldrich) and coverslips mounted on slides.

## Immunofluorescence

Fibroblasts were grown on sterilized glass coverslips coated with 50 µg/ml fibronectin (F1141, Sigma-Aldrich) and fixed with 4% paraformaldehyde in PBS for 20 min. Following fixation, cells were rinsed in PBS, permeabilized in PBS + 0.3% Triton X-100 for 7 min, and then blocked in 10% FBS + PBS for 1 h. Both, primary and secondary antibodies were diluted in PBS + 0.05% Tween-20 containing 5% FBS as follows. Primary antibodies: rabbit anti-53BP1 (4937, Cell Signaling Technology), 1:100; mouse anti-p21 (SC-6246, Santa Cruz Biotechnology), 1:800; mouse anti-Aurora B (Aim-1; 611082, BD Biosciences), 1:500; rabbit anti-cGAS (15102, Cell Signaling Technology), 1:200; mouse anti-Hec1 (ab3613, Abcam), 1:1,500; mouse anti-Plk1 (SC-17783, Santa Cruz Biotechnology), 1:2,000; rabbit anti-MCAK [49], 1:5,000; mouse anti-α-Tubulin (T5168, Sigma-Aldrich), 1:1,500; human anti-centromere antibody (ACA; kindly provided by Dr. W. C. Earnshaw), 1:3,000; rabbit anti-Aurora B phospho T232 (600-401-677, ROCKLAND), 1:1,000; mouse anti-Retinoblastoma (554136, BD Biosciences), 1:100. Secondary antibodies: Alexa Fluor-488, Alexa Fluor-568 and Alexa Fluor-647 (Life Technologies), all 1:1,500. DNA was counterstained with 0.5 µg/ml DAPI (Sigma-Aldrich) and coverslips mounted on slides with proper mounting solution.

## Fluorescence dissipation after photoconversion

Human dermal fibroblasts of neonatal and elderly cells transduced with inducible mEOS-Tubulin were cultured for 24–48 h in MEM (without phenol red) + 750 ng/ml doxycycline on fibronectin-coated glass coverslips. 5 µM MG-132 was added prior to rose

chamber assembly to prevent mitotic exit. During imaging, cells were maintained at 37°C using a heated stage. Images were acquired with a Plan Apo VC 60×, 1.4 NA, oil immersion objective (Nikon) using a Quorum WaveFX-X1 spinning-disk confocal system on a Nikon Eclipse Ti microscope, equipped with an ILE laser source (Andor Technology), a Mosaic digital mirror (Andor Technology), and a Photometrics Evolve 512 Delta camera. Metaphase cells were identified by differential interference contrast (DIC) microscopy. Photoconversion was performed with a 405 nm laser (20% power, 500 ms pulse) on a rectangular region of interest over one half of the mitotic spindle. Fluorescence *z*-stacks with a 1 μm step size (three slices) were captured every 15 s for 4 min. Red fluorescence dissipation after photoconversion was quantified from maximum intensity projections using MetaMorph® software (Molecular Devices). Average pixel intensities were measured within an area surrounding the region of highest fluorescence intensity, and background subtraction was performed using an equally sized area from the non-activated half-spindle at each time point. Fluorescence intensities were corrected for photobleaching using values of fluorescence loss obtained from photoconverted 1 μM Taxol stabilized spindles. Fluorescence intensities were then normalized to the first time point after photoconversion for each cell. To measure k-MT stability, the average fluorescence intensity at each time point was fit to a two-phase exponential decay curve $[F(t) = A_1 e^{-k_1 t} + A_2 e^{-k_2 t}]$ using GraphPad software, where $A_2$ is the percentage of photoconverted fluorescence attributable to the slow decay process with a decay rate of $k_2$ [28]. k-MT half-life $(t^{1/2})$ in minutes was calculated as $\ln 2/k_2$. For high stringency, we only considered curves with good fit ($R^2 \geq 0.95$).

### Phase-contrast live-cell imaging

Fibroblasts grown in ibiTreat polymer-coated μ-slide (Ibidi GmbH, Germany) were imaged using a Zeiss Axiovert 200M inverted microscope (Carl Zeiss, Oberkochen, Germany) equipped with a CoolSnap camera (Photometrics, Tucson, USA), XY motorized stage and NanoPiezo Z stage, under controlled temperature, atmosphere, and humidity. Neighbor fields (20–25) were imaged every 2.5 min for 24–48 h, using a 20×/0.3 NA Aplan objective. The "Stitch Grid" (Stephan Preibisch) plug-in from ImageJ/Fiji software was used to stitch neighboring fields for image analysis.

### Spinning-disk confocal microscopy

Fibroblasts were grown in ibiTreat polymer-coated 35-mm μ-dishes (Ibidi GmbH, Germany) and imaged using the Andor Revolution XD spinning-disk confocal system (Andor Technology, Belfast, UK), equipped with an electron-multiplying charge-coupled device iXonEM Camera and a Yokogawa CSU 22 unit based on an Olympus IX81 inverted microscope (Olympus, Southend-on-Sea, UK). The system was driven by Andor IQ software, and laser lines at 488 and 561 nm were used for excitation of GFP and mCherry, respectively. *Z*-stacks (0.8–1.0 μm) covering the entire volume of individual mitotic cells were collected every 1.5 min using a PlanApo 60×/1.4 NA objective. ImageJ/Fiji software was used to edit the movies in which every image represents a maximum intensity projection of all *z*-planes.

### Fluorescence microscopy

Cells with calcium-stabilized k-fibers or stained for specific kineto-chore/centromere-bound proteins (Aurora B/Aim-1, Hec1/Ndc80, MCAK, Plk1, pAuroraB T232, and ACA) were imaged using a Zeiss AxioImager Z1 (Carl Zeiss, Oberkochen, Germany) motorized upright epifluorescence microscope, equipped with an Axiocam MR camera, and operated by the Zeiss Axiovision v4.7 software. *Z*-stacks (0.24 μm) covering the entire volume of individual mitotic cells were collected using a PlanApo 63×/1.40 NA objective. Image deconvolution was performed with the AutoQuant X2 software (Media Cybernetics).

### Automated microscopy

For FISH experiments, MN counts, cGAS immunofluorescence, and SA biomarkers, images were captured with the IN Cell Analyzer 2000 (GE Healthcare, UK) equipped with a Photometrics CoolSNAP K4 camera and driven by the GE IN Cell Analyzer 2000 v5.2 software, using a Nikon 20×/0.45 NA Plan Fluor objective and a Nikon 40×/0.95 NA Plan Fluor objective, respectively.

### Image analysis

Both, live-cell phenotypes (mitotic duration, lagging chromosomes) and fixed-cell experiments (protein intensity, k-fiber intensity, KT distances, FISH, MN counts, mitotic stages, cGAS positivity, and SA biomarkers) were blindly quantified using ImageJ/Fiji software. For the analysis of protein intensity levels, the kinetochore area was taken into consideration and Aurora B/Aim-1, Hec1/Ndc80, MCAK, Plk1, and pAuroraB T232 levels were then corrected for the background and normalized to ACA levels (also corrected for the background). For analysis of calcium-stable k-fibers, α-Tubulin intensity levels were normalized for the mitotic spindle area of each individual cell and background-corrected. For MN frequencies, interphase cells with DNA aggregates separate from the primary nucleus were considered, while interphase cells with an apoptotic appearance were excluded. DNA aggregates co-localizing with cGAS and/or Retinoblastoma (Rb) were scored as MN positive for cGAS or Rb, respectively. Intact MN were cGAS⁻/Rb⁺, while disrupted MN were cGAS⁺/Rb⁻. For the analysis of SA biomarkers (53bp1/p21 and SA-β-galactosidase), fluorescence intensity thresholds were set and used consistently for all samples within each experiment. In case of SA-β-galactosidase activity, only cells displaying > 5 fluorescent granules were considered positive.

### Western blot

Both asynchronous and mitotic cell populations were analyzed by Western blot. Mitotic cell populations were collected by shake-off of cell culture flasks enriched for Mitotic Index following a 16-h treatment with STLC. Cell pellets were resuspended in lysis buffer (150 nM NaCl, 10 nM Tris–HCl pH 7.4, 1 nM EDTA, 1 nM EGTA, 0.5% IGEPAL) with protease inhibitors. Protein content was determined using the Lowry Method (DC™ Protein Assay, Bio-Rad) according to the manufacturer's instructions. 20 μg of extract was then loaded for SDS-polyacrylamide gel electrophoresis and transferred onto nitrocellulose membranes for Western blot analysis.

Non-specific sites were blocked with TBS-T (50 mM Tris–HCl, pH = 7.4, 150 mM NaCl, 0.05% Tween-20) supplemented with 5% non-fat dry milk for 1 h. Both, primary and secondary antibodies were diluted in TBS-T containing 2% non-fat milk as follows. Primary antibodies: mouse anti-Aurora B (Aim-1; 611082, BD Biosciences), 1:250; mouse anti-GFP (46-0092, Invitrogen), 1:2,000; rabbit anti-cGAS (15102, Cell Signaling Technology), 1:100; mouse anti-Hec1 (SC-515550, Santa Cruz Biotechnology), 1:500; rabbit anti-MCAK [49], 1:5,000; mouse anti-Plk1 (SC-17783, Santa Cruz Biotechnology), 1:1,000; mouse anti-α-Tubulin (T5168, Sigma-Aldrich), 1:100,000; and mouse anti-GAPDH (60004, ProteinTech), 1:50,000. Secondary antibodies: horseradish peroxidase (HRP)-conjugated goat anti-rabbit (SC-2004, Santa Cruz Biotechnology) and goat anti-mouse (SC-2005, Santa Cruz Biotechnology), both at 1:3,000. HRP conjugates were detected using Clarity Western ECL Substrate reagent (Bio-Rad Laboratories) according to manufacturer's instructions. A GS-800 calibrated densitometer operated by the Quantity one I-D Analysis Software v4.6 (Bio-Rad Laboratories) was used for quantitative analysis of protein levels.

### Cytokine array

Cell medium supernatants from neonatal and octogenarian HDFs for 24 h in the absence of FBS [50] were harvested and centrifuged for 5 min at $200\ g$ to remove dead cells and cell debris. Levels of secreted cytokines were then analyzed using the Proteome Profiler Human Cytokine Array Kit (R&D SYSTEMS) according to the manufacturer's instructions. Quantitative analysis of dot blots was performed on a GS-800 calibrated densitometer operated by the Quantity one I-D Analysis Software v4.6 (Bio-Rad Laboratories).

### Quantitative PCR

Total RNA from both asynchronous and mitotic cell populations was extracted using RNeasy® Mini Kit (Qiagen). 1 µg of total RNA was reverse-transcribed using the iScript™ cDNA synthesis kit (Bio-Rad Laboratories). qPCR was performed using iTaq™ Universal SYBR® Green Supermix in a CFX96/384 Touch™ Real-Time PCR Detection System and analyzed using the CFX Maestro Software (all from Bio-Rad Laboratories). Primers used are listed in Appendix Table S1.

### Targeted transcriptome sequencing and Bioinformatics

Total RNA was extracted as described above. 10 ng was reverse-transcribed using the AmpliSeq Whole Transcriptome primers with the included SuperScript VILO cDNA Synthesis kit (Thermo Fisher Scientific). cDNA was used for target amplification (12 cycles) with Ion AmpliSeq primers and technology. Barcoded adapters were added and ligated to individual reactions according to the instructions of Ion AmpliSeq™ Transcriptome Human Gene Expression Kit (Thermo Fisher Scientific). The pooled libraries were processed on Ion Chef™ System, and the resulting 550™ chip was sequenced on a Ion S5™ XL System (all from Thermo Fisher Scientific). Data were processed with the Ion Torrent platform-specific pipeline software Torrent Suite v5.8.0 to generate sequence reads, trim adapter sequences, filter and remove poor signal reads, and split the reads according to the barcode. FASTQ and/or BAM files generated with the Torrent Suit plug-in FileExporter v5.10.0.0 were analyzed using Torrent Suite™ v5.8.0 Software (Thermo Fisher Scientific) running Ion AmpliSeq™ RNA plug-in v5.10.1.2, coverageAnalysis plug-in v5.10.0.3, and target region hg19_AmpliSeq_Transcriptome_21 K_v1. Differential gene expression analysis and principal component analysis (PCA) were performed using the Transcriptome Analysis Console v4.0.2 (TAC) Software (Thermo Fisher Scientific). Gene expression was defined as significantly different based on $P < 0.05$ and fold change cutoff value $< -1.6$ or $> 1.6$ for the comparison between neonatal (HDFN) and elderly (HDF87y) empty vector-transduced (control) cultures (Dataset EV1). Heatmaps were generated using the transcript counts normalized against the sample library size, which were then scaled by gene with normalized scores or $z$-scores (*i.e.*, a value of 0 refers to the mean gene expression of that gene across all libraries; $> 0$ or + indicates expression above, and $< 0$ or – indicates expression under this mean gene expression). Hierarchical clustering was performed using the R library heatmap.2. function with R package v3.6.1. Targeted transcriptome sequencing data represent two independent experimental replicates of each biological sample.

### Statistical analysis

All experiments were repeated at least two times unless otherwise stated. Sample sizes and statistical tests used for each experiment are indicated in the respective figure captions. Data are shown as mean ± SD or mean ± SEM as indicated. GraphPad Prism version 7 was used to analyze all the data. Data were tested for parametric versus non-parametric distribution using D'Agostino–Pearson omnibus normality test. Two-tailed Mann–Whitney or chi-square tests were then applied accordingly to determine the statistical differences between different groups (*$P < 0.05$, **$P < 0.01$, ***$P < 0.001$, ****$P < 0.0001$, and NS not significant $P > 0.05$).

# Data availability

The targeted transcriptome analysis data generated for this study have been deposited at the ArrayExpress database at EMBL-EBI (www.ebi.ac.uk/arrayexpress) under accession number E-MTAB-8634.

**Expanded View** for this article is available online.

### Acknowledgements

We would like to thank Bernardo Orr for discussions and Benjamin Kwok for kindly providing UMK57. We also thank Patrícia Oliveira for assistance with targeted transcriptome analysis, and laboratory members for feedback on the manuscript and throughout this project. E.L. holds an FCT Investigator Post-doctoral Grant (IF/00916/2014), and M.V. holds an FCT Fellowship SFRH/BD/133478/2017 from FCT/MCTES (Fundação para a Ciência e a Tecnologia/Ministério da Ciência, Tecnologia e Ensino Superior). The following project grants supported this work: FEDER - Fundo Europeu de Desenvolvimento Regional funds through the COMPETE 2020 - Operational Program for Competitiveness and Internationalization (POCI), Portugal 2020, and by Portuguese funds through FCT/MCTES in the framework of the project POCI-01-0145-FEDER-031120 (PTDC/BIA-CEL/31120/2017). NORTE-01-0145-FEDER-000029 funded by North Regional Operational Program (NORTE2020) under PORTUGAL2020

Partnership Agreement through FEDER; National Funds through FCT under the project PTDC/BEX-BCM/2090/2014; POCI-01-0145-FEDER-007274 i3S framework project co-funded by COMPETE 2020/PORTUGAL 2020 through European Regional Development Fund (FEDER) and by FCT. The authors acknowledge the support of the i3S Scientific Platforms ALM and BS, members of the PPBI (PPBI-POCI-01-0145-FEDER-022122).

## Author contributions

MB-V, JCM, and MR generated resources and performed and analyzed experiments. JDW performed the photoconversion experiments and analyzed the data. EL conceived and supervised the project, designed the experiments, and analyzed the data. MB-V and EL wrote the manuscript with input from JW and DC.

## Conflict of interest

The authors declare that they have no conflict of interest.

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
