## [Review Process File · EMBO Reports]

Small molecule inhibition of aging-associated chromosomal instability delays cellular senescence

Monika Barroso-Vilares, Joana C. Macedo, Marta Reis, Jessica D. Warren, Duane Compton, Elsa Logarinho

Review timeline:

Submission date:	9 September 2019
Editorial Decision:	7 October 2019
Revision received:	20 December 2019
Editorial Decision:	4 February 2020
Revision received:	5 February 2020
Accepted:	13 February 2020

Transaction Report:

1st Editorial Decision

7 October 2019

Thank you for the submission of your research manuscript to our journal. We have now received the full set of referee reports that is copied below.

As you will see, the referees acknowledge that the findings are potentially interesting but they also point out several technical concerns and have a number of suggestions for how the study should be strengthened, which should be addressed in a revision.

Given the very constructive comments, we would like to invite you to revise your manuscript with the understanding that the referee concerns (as detailed in their reports) must be fully addressed and their suggestions taken on board. Please address all referee concerns in a complete point-by-point response. Acceptance of the manuscript will depend on a positive outcome of a second round of review. It is EMBO reports policy to allow a single round of revision only and acceptance or rejection of the manuscript will therefore depend on the completeness of your responses included in the next, final version of the manuscript.

Revised manuscripts should be submitted within three months of a request for revision; they will otherwise be treated as new submissions. Please contact us if a 3-months time frame is not sufficient for the revisions so that we can discuss the revisions further.

2) individual production quality figure files as .eps, .tif, .jpg (one file per figure). Please download our Figure Preparation Guidelines (figure preparation pdf) from our Author

Guidelines pages

<https://www.embopress.org/page/journal/14693178/authorguide> for more info on how to prepare your figures.

4) a complete author checklist, which you can download from our author guidelines (<<https://www.embopress.org/page/journal/14693178/authorguide>>). Please insert information in the checklist that is also reflected in the manuscript. The completed author checklist will also be part of the RPF.

5) Please note that all corresponding authors are required to supply an ORCID ID for their name upon submission of a revised manuscript (<<https://orcid.org/>>). Please find instructions on how to link your ORCID ID to your account in our manuscript tracking system in our Author guidelines (<<https://www.embopress.org/page/journal/14693178/authorguide#authorshipguidelines>>)

6) Supplementary Information:

We have three levels: (1) Expanded View (EV) figures, (2) a single pdf called Appendix, (3) Datasets.

- Expanded View (EV) Figures and Tables are collapsible/expandable online.

A maximum of 5 EV Figures can be typeset. EV Figures should be cited as 'Figure EV1, Figure EV2' etc.. in the text and their respective legends should be included in the main text after the legends of regular figures.

=> In practical terms this means that the header "Supplemental Information" should be changed to "Expanded view figure legends" and the figure titles should be changed to "Figure EV1, Figure EV2 etc).

- Appendix: Supplementary Table 1 will be part of the Appendix and should be called "Appendix Table S1". In case the revision results in more than five EV figures, all additional figures can also be part of the Appendix.

The Appendix is a single PDF file, which starts with a short Table of Content including page numbers. Please note that even if the Appendix contains only one figure or one table, we need a title page. Appendix figures or tables should be referred to in the main text as: "Appendix Figure S1, Appendix Figure S2", "Appendix table S1" etc. See detailed instructions regarding expanded view here:

<<https://www.embopress.org/page/journal/14693178/authorguide#expandedview>>

- Additional Tables/Datasets should be labeled and referred to as Table EV1, Dataset EV1, etc.

Legends have to be provided in a separate tab in case of .xls files. Alternatively, the legend can be supplied as a separate text file (README) and zipped together with the Table/Dataset file.

=> this applies to your Data/Table S1 - S4. The nomenclature should be changed to "Dataset EV1 - EV4".

7) Data availability: Before submitting your revision, primary datasets produced in this study need to be deposited in an appropriate public database (see

<<https://www.embopress.org/page/journal/14693178/authorguide#datadeposition>>).

Specifically, we would kindly ask you to provide public access to the RNA seq datasets.

The accession numbers and database should be listed in a formal "Data Availability " section (placed after Materials & Method) that follows the model below (see also

<<https://www.embopress.org/page/journal/14693178/authorguide#datadeposition>>). Please note that

the Data Availability Section is restricted to new primary data that are part of this study.

Data availability

8) We would also encourage you to include the source data for figure panels that show essential data.

Numerical data should be provided as individual .xls or .csv files (including a tab describing the data). For blots or microscopy, uncropped images should be submitted (using a zip archive if multiple images need to be supplied for one panel). Additional information on source data and instruction on how to label the files are available
<<https://www.embopress.org/page/journal/14693178/authorguide#sourcedata>>.

9) Our journal encourages inclusion of **data citations in the reference list** to directly cite datasets that were re-used and obtained from public databases. Data citations in the article text are distinct from normal bibliographical citations and should directly link to the database records from which the data can be accessed. In the main text, data citations are formatted as follows: "Data ref: Smith et al, 2001" or "Data ref: NCBI Sequence Read Archive PRJNA342805, 2017". In the Reference list, data citations must be labeled with "[DATASET]". A data reference must provide the database name, accession number/identifiers and a resolvable link to the landing page from which the data can be accessed at the end of the reference. Further instructions are available at
<<https://www.embopress.org/page/journal/14693178/authorguide#referencesformat>>.

10) Regarding data quantification:

- Please ensure to specify the name of the statistical test used to generate error bars and P values, the number (n) of independent experiments underlying each data point (not replicate measures of one sample), and the test used to calculate p-values in each figure legend. Discussion of statistical methodology can be reported in the materials and methods section, but figure legends should contain a basic description of n, P and the test applied.
- IMPORTANT: Please note that error bars and statistical comparisons may only be applied to data obtained from at least three independent biological replicates. If the data rely on a smaller number of replicates, scatter blots showing individual data points are recommended.
- Graphs must include a description of the bars and the error bars (s.d., s.e.m.).
 - Please also include scale bars in all microscopy images.

11) As part of the EMBO publication's Transparent Editorial Process, EMBO reports publishes online a Review Process File to accompany accepted manuscripts. This File will be published in conjunction with your paper and will include the referee reports, your point-by-point response and all pertinent correspondence relating to the manuscript.

I look forward to seeing a revised version of your manuscript when it is ready. Please let me know if you have questions or comments regarding the revision.

REFEREE REPORTS

Referee #1:

The manuscript by Barroso-Vilares et al. provides a comprehensive characterization of the molecular mechanisms underlying age-associated chromosomal instability. The authors nicely show that aged cells display defective kinetochore-microtubules interactions and overexpression of Kif2b or MCAK is able to improve mitotic fidelity, delaying the onset of senescence. Interestingly, they also found that aged cells treated with a chemical compound able to potentiate the activity of MCAK is able to suppress chromosomal instability in aged cells, with important implications for human health. The paper is well written, significant, and the main conclusions are compelling and novel. Thus, the paper is suitable for publication in EMBO Reports after minor revisions.

I have a few comments below.

1. Although the assay employing ZM447439 is showing that elderly cells have a higher frequency of chromosome segregation errors, it would be important to test this also in an orthogonal experimental set-up, such as inhibition of Mps1. This would definitely strengthen the conclusions.
2. It is not clear to this reviewer whether or not MNs in aged cells have nuclear rupture. One could infer this from cGAS staining, but it would be much better to have a staining for Rb, Lsd1 or introduce a GFP-NLS construct.
3. The authors found that aged cells show a SASP signature. It would interesting to test whether this also leads to the secretion of cytokines and inflammatory modulators. This experiment would be essential in order to add a mechanistic link between aging-associated CIN and senescence.

Referee #2:

This is a well written and interesting manuscript by Barroso-Vilares et al. The major conclusion posits age-associated CIN is not merely a "side-effect" of the aging process but rather a driver of aging itself by contributing to senescence pathways. The authors present convincing evidence that older primary dermal fibroblasts have more kinetochore MT attachments (k-MTs) than younger cells and, perhaps as a result of increased k-MT number, that the older cells do not correct erroneous attachments as efficiently as younger cells, which leads to CIN. Interestingly, a number of KT/centromere associated k-MT regulators are present at reduced levels in the older cells as well. The authors focus on the contribution of MCAK to age-associated CIN and found that enhancement of its activity by either over-expression or small molecule enhancer called UMK57 rescues age-associated CIN and, importantly, the senescence-related markers that were assessed.

I found this manuscript to be technically well done, well-written, and interesting. While I feel that the feasibility of targeting MCAK as a general aging treatment may be somewhat oversold by the authors because 1) the high concentration (1 micromolar - see comments below) necessary to achieve the rescue may yield off-target effects and 2) globally inhibiting this important catastrophe-factor in an organism may lead to significantly worse side-effects than aging itself, I acknowledge that it's an exciting idea that warrants further investigation. I have comments that I would like the authors to remedy prior to publication.

Comments:

- 1) The authors focus on MCAK in this manuscript yet their data on MCAK levels (and other regulators) in a supplemental figure. The MCAK RNA, protein, and KT/centromere level quantifications should be added to figure 1.
- 2) How does UMK57 work? Does it affect MCAK protein levels in cells at the concentration it is used? On the concentration front, in the prior work (Orr, Talje et al.) the optimal concentration for suppressing CIN was 100 nM. Why is 10X that amount (1 micromolar) required here? Do the authors think it is solely cell-type specific? If so, then this wouldn't bode well for a treatment as each cell-type in an organism will respond in very different ways to different concentrations.
- 3) On a related note, the concentration that is used is based on timing from NEB-anaphase onset, but

the relevant metrics here are 1) reduced CIN, 2) reduced incidence of micronuclei, and reduced/delayed cellular senescence. How low a concentration could you go to observe these relevant impacts?

4) It was unclear from the methods if BOTH the neonatal and octogenarian cell lines are from male donors. If so, great. If not, could sex be a biological variable here? Has sex-linked differences been examined with regards to k-MT number and k-MT regulators?

5) The HDF N and HDF 87y cells are not labeled in Fig. 1E. I presume the 87F is the one with the lagging chromosome but it should be labeled as such. If it's simply showing examples of each condition in the same aged cell then the figure legend is mis-leading as it states that it is live-cell imaging of each cell type.

6) In terms of the error correction assay data presented in Figure 1E, there is something to be said about the fact that both cell lines look to have a comparable fold-increase in lagging chromosomes compared to the -ZM condition. This makes the ZM data less informative as the baseline level of lagging chromosome following STLC WO is ~2 fold higher in the 87y cells compared to the Ns.

7) "Top" and "bottom" appear to be used incorrectly in figure legend 1G.

8) What is responsible for the increase in time from NEB-anaphase in the old cells? Does it take longer for the cells to align their chromosomes or is the duration of metaphase longer?

9) The presentation of the transcriptomics results in MCAK and Kif2b over-expressing cells was somewhat mis-leading in the text. For example, 17 (out of 20 SASP genes) were "correctly altered following over-expression of kinesin-13 proteins" gave me the impression that the transcript levels were restored to HDF N levels. Instead they moved in the right direction but were not fully "corrected". Please phrase this similarly to how it is presented for the cGAS, etc. genes where it is clearly stated that MCAK/Kif2b over-expression gave a partial rescue.

10) With regards to the adaptation to UMK57, while I agree it is not substantial the data presented in Figure 5H shows that there is a reduction ($p < 0.05$) in centromeric pAurB levels after 96 hours in UMK57. The stats are not shown in the plot, but are the levels after 96h also reduced relative to the 24 hour condition? The text in the main body of the manuscript should acknowledge this reduction as it is presently reads that "analysis.. at centromeres did not reveal substantially reduced levels in.. metaphase (Fig 5G,H) after 96 hours."

Referee #3:

Logarinho and colleagues present a nice follow up to the Logarinho laboratory's previous study demonstrating that aged human fibroblasts exhibit chromosomal instability and aneuploidy, leading to senescence. This present study nicely advances these findings by demonstrating that mis-regulated mitotic proteins leading to incorrect microtubule dynamics are likely a cause of the elevated chromosome mis-segregation seen in elderly cells, since modulation of microtubule dynamics can rescue these defects.

The authors use two independent methods to correct aberrant kinetochore-microtubule attachments, taking advantage of a recently discovered small molecule potentiator of the microtubule depolymerase MCAK, and confirming these results with overexpression of GFP-tagged MCAK. Interestingly they also go on to show that the rescue of mitotic defects, aneuploidy and senescence can be long-lived (above 96 hours) in contrast to cancer cells that appear to circumvent the rescue of chromosomal instability within a short time period, as previously shown by the Compton laboratory. Overall the experimental design is well conceived, the experiments well performed and clearly presented. The manuscript is well written and clear. I recommend this for publication in EMBO reports, and have only a couple of minor points/suggestions:

1. Figure 1C and 1D: what does n=50 refer to, individual inter-KT distance measurements, or cells? If individual KT measurements could they provide the number of total cells analysed too?
2. Figure 2A: GFP-MCAK band in the Western blot looks very odd in the 87y fibroblasts, why is this? Could the authors provide an immunofluorescence image of the GFP-MCAK cells?
3. Figure 3A and 3B: It would be helpful to include an example image of the 53BP1/p21 and SA- β -Gal IF (shown nicely in the supplementary data) in the main figure.

Point-by-point responses to Reviewers

Referee #1:

The manuscript by Barroso-Vilares et al. provides a comprehensive characterization of the molecular mechanisms underlying age-associated chromosomal instability. The authors nicely show that aged cells display defective kinetochore-microtubules interactions and overexpression of Kif2b or MCAK is able to improve mitotic fidelity, delaying the onset of senescence. Interestingly, they also found that aged cells treated with a chemical compound able to potentiate the activity of MCAK is able to suppress chromosomal instability in aged cells, with important implications for human health. The paper is well written, significant, and the main conclusions are compelling and novel. Thus, the paper is suitable for publication in EMBO Reports after minor revisions.

We thank the reviewer for his/her positive feedback and for underlining the significance and novelty of our findings.

I have a few comments below.

We have addressed all the points raised by the reviewer as specified below.

1. Although the assay employing ZM447439 is showing that elderly cells have a higher frequency of chromosome segregation errors, it would be important to test this also in an orthogonal experimental set-up, such as inhibition of Mps1. This would definitely strengthen the conclusions.

We followed the Reviewer's suggestion and included an experimental set-up with Mps1 inhibitor (Appendix Figure S1).

In our previous study (*Macedo JM et al., Nat Communics 2018*) we treated young and elderly cells with 5 μ M Mps1 inhibitor (AZ3146, TOCRIS, USA), a concentration that inhibits the SAC, which allowed us to conclude that the mitotic delay in elderly cells is due to defective K-MT attachments activating the SAC.

In this study, we used Mps1 inhibitor at nanomolar concentration (500 nM), so that SAC is weakened but not totally inhibited. Under this condition, we found higher frequency of chromosome mis-segregation in elderly cells, strengthening our previous results from the assay employing ZM447438 (lines 133-138, page 5). The data are shown in Appendix Figure S1.

2. It is not clear to this reviewer whether or not MNs in aged cells have nuclear rupture. One could infer this from cGAS staining, but it would be much better to have a staining for Rb, Lsd1 or introduce a GFP-NLS construct.

We have now included double immunostaining analysis of cGAS and Rb in MNs. As expected, cGAS+ MNs were Rb-, whereas cGAS- MNs were Rb+. Our data show that aged cells have higher frequency of MNs staining cGAS+/Rb-, thus demonstrating that these MNs more often have nuclear rupture (lines 185-188, page 7) (revised Fig EV3C,D). Nevertheless, aged cells also have higher frequency of MN staining cGAS-/Rb+, which likely represent transiently intact MN before getting disrupted.

Importantly, UMK57 treatment was able to decrease the frequency of MNs, both those staining cGAS+/Rb- and cGAS-/Rb+ (revised Fig EV5B). Noteworthy, cGAS siRNA-depletion in aged cells decreased the frequency of MNs staining cGAS+ but without reducing the total frequency of MNs (Fig EV3F), which further supports that UMK is specifically modulating MN frequency.

3. The authors found that aged cells show a SASP signature. It would be interesting to test whether this also leads to the secretion of cytokines and inflammatory modulators. This experiment would be essential in order to add a mechanistic link between aging-associated CIN and senescence.

We acknowledge the Reviewer's suggestion and we have now included data demonstrating that elderly cells, besides their autocrine SASP signature, also secrete inflammatory cytokines into the medium as determined using the R&D systems cytokine assay. This supports the mechanistic link between aging-associated CIN and senescence.

These data are shown in Figure EV3A of the revised version of the manuscript (lines 178-179, page 7).

Referee #2:

This is a well written and interesting manuscript by Barroso-Vilares et al. The major conclusion posits age-associated CIN is not merely a "side-effect" of the aging process but rather a driver of aging itself by contributing to senescence pathways. The authors present convincing evidence that older primary dermal fibroblasts have more kinetochore MT attachments (k-MTs) than younger cells and, perhaps as a result of increased k-MT number, that the older cells do not correct erroneous attachments as efficiently as younger cells, which leads to CIN. Interestingly, a number of KT/centromere associated k-MT regulators are present at reduced levels in the older cells as well. The authors focus on the contribution of MCAK to age-associated CIN and found that enhancement of its activity by either over-expression or small molecule enhancer called UMK57 rescues age-associated CIN and, importantly, the senescence-related markers that were assessed. I found this manuscript to be technically well done, well-written, and interesting.

We thank the Reviewer for his/her positive comments.

While I feel that the feasibility of targeting MCAK as a general aging treatment may be somewhat oversold by the authors because 1) the high concentration (1 micromolar - see comments below) necessary to achieve the rescue may yield off-target effects and 2) globally inhibiting this important catastrophe-factor in an organism may lead to significantly worse side-effects than aging itself, I acknowledge that it's an exciting idea that warrants further investigation.

We value the Reviewer concern and we have addressed this issue in order to improve the quality of the manuscript and reinforce the idea that targeting MCAK to delay senescence (hence, promoting healthy aging) could be feasible without side-effects.

I have comments that I would like the authors to remedy prior to publication.

We have addressed all the points raised by the Reviewer as specified below.

Comments:

1) The authors focus on MCAK in this manuscript yet their data on MCAK levels (and other regulators) in a supplemental figure. The MCAK RNA, protein, and KT/centromere level quantifications should be added to figure 1.

We agree with the Reviewer suggestion and we have now included data on MCAK levels in a revised version of Figure 1, namely Fig.1I and J, whilst maintaining all prior data in Figure EV2.

2) How does UMK57 work?

A high-throughput screen was performed to identify small molecules that modulate the activities of kinesin-13 proteins (Talje et al., 2014). This screen identified a kinesin-13 inhibitor that was previously reported (Talje et al., 2014), but also came out with a family of compounds that potentiate the microtubule depolymerizing activity of kinesin-13 proteins *in vitro*. Complete characterization of how these compounds (UMK57 included) affect the biochemistry of kinesin-13 proteins *in vitro* will be provided elsewhere, through in depth single molecule analysis and X-ray crystallography, that we are certain the Reviewer understands is beyond the scope of the present work. Nevertheless, as reported in Orr B et al. 2017, we know that UMK57 displays exquisite selectivity toward MCAK as shown in Figure S1 of that manuscript.

Does it affect MCAK protein levels in cells at the concentration it is used?

Orr B et al. have previously shown (Figure 1C,D in their reference) that MCAK protein levels do not change in U2OS cells treated with 100nM UMK57. In the revised version of the manuscript, we now show that 1µM of UMK57 does not overtly change MCAK protein levels, neither in young nor in elderly cells (Figure EV4E).

On the concentration front, in the prior work (Orr, Talje et al.) the optimal concentration for suppressing CIN was 100 nM. Why is 10X that amount (1 micromolar) required here? Do the authors think it is solely cell-type specific? If so, then this wouldn't bode well for a treatment as each cell-type in an organism will respond in very different ways to different concentrations.

We appreciated that the Reviewer raised this question. By answering this, we ended up strengthening our results that point to a completely new direction for how to consider influencing cellular aging. The Reviewer certainly acknowledges that the results nevertheless represent a preclinical proof-of-concept and not a clinical assessment of the use of UMK57 in human patients, which would need substantial refinement.

We have titrated the optimal concentration of UMK57 based on more relevant metrics as requested by the Reviewer in point 3 below. The concentration of UMK57 needed to significantly rescue the aging-associated phenotypes was 1.0 μ M (0.1 μ M was insufficient). The titration data are now shown in Figure EV4A-D. In Figure EV4A we show that 1.0 μ M UMK57 treatment of neonatal and elderly cells does not increase the percentage of cells in prometaphase as expected if k-MT attachments were too unstable. In Figure EV4B,C we show that 1.0 μ M UMK57 treatment is needed to significantly rescue chromosome mis-segregation rate and micronuclei in elderly cells, respectively. Also, in Figure EV4D we show that SA- β -gal activity is only significantly rescued in elderly cell cultures treated with 0.5-1.0 μ M UMK57.

We consider this 10X amount required to suppress CIN to be in agreement with cell-type specific levels of MCAK. As previously reported by Orr B et al., 100nM UMK57 treatment was excessive in cells overexpressing MCAK (Figure 1F in this reference). Thus, one should expect a higher concentration of UMK57 to be required in cells with lower MCAK levels (e.g. elderly fibroblasts). To validate this rationale, we have included in the revised version of the manuscript, a comparative analysis of MCAK protein levels in neonatal fibroblasts vs. fibrosarcoma HT1080 cell line (Figure EV4F). Neonatal fibroblasts have 37% MCAK levels in comparison to HT1080 cancer cells (Figure EV4F), and elderly fibroblasts have 40% MCAK levels in comparison to neonatal fibroblasts (Figure EV2B). Moreover, we show that higher MCAK levels in HT1080 cells turn them more sensitive to UMK57. Concentrations of UMK57 \geq 0.25 μ M already generate over unstable K-MT attachments (Figure EV4G-I). In contrast, 1 μ M UMK57 is needed to reduce CIN in elderly fibroblasts (which have the lowest levels of MCAK). Importantly, we show that 1 μ M UMK57 treatment does not affect young cells, neither at 24 hrs (Figure 4A-H), nor at prolonged treatment periods (96 hrs or 4 weeks; Figures 5A-H and EV5A-I). This supports the safety use of UMK57 in the context of aging, which is different from the context of cancer where cells typically overexpress MCAK. Indeed, Orr B et al. have also found that whereas non-transformed RPE1 cells required higher doses of UMK57 (\geq 1 μ M) before an effect on mitotic duration/cell proliferation was observed, U2OS cancer cells were affected by lower doses (\geq 0.1 μ M) (Figure S2H,I in this reference).

3) On a related note, the concentration that is used is based on timing from NEB-anaphase onset, but the relevant metrics here are 1) reduced CIN, 2) reduced incidence of micronuclei, and reduced/delayed cellular senescence. How low a concentration could you go to observe these relevant impacts?

We have titrated the optimal concentration of UMK57 based on more relevant metrics. The concentration of UMK57 needed is indeed 1.0 μ M, as lower concentrations (0.1, 0.25 or 0.5 μ M) turned out to be insufficient to significantly rescue the aging-associated phenotypes. The titration data are now shown in Figure EV4A-D.

4) It was unclear from the methods if BOTH the neonatal and octogenarian cell lines are from male donors. If so, great.

Yes, both neonatal and octogenarian dermal fibroblasts were retrieved from male donors.

If not, could sex be a biological variable here? Has sex-linked differences been examined with regards to k-MT number and k-MT regulators?

In our previous study, besides having used HDFs derived from male donors as in this study, we also used female mouse adult fibroblasts, for both of which we reported similar age-associated mitotic phenotypes (Macedo et al., *Nat Commun* 2018, *Supplementary Figure 2*). In this study, sex-linked differences have not been examined with regards to k-MT number and k-MT regulators. Also, *MCAK* and *KIF2B* genes are not encoded on the X chromosome.

5) The HDF N and HDF 87y cells are not labeled in Fig. 1E. I presume the 87F is the one with the lagging chromosome but it should be labeled as such. If it's simply showing examples of each condition in the same aged cell then the figure legend is mis-leading as it states that it is live-cell imaging of each cell type.

It was live-cell imaging of each cell type. We have now labeled Fig. 1E panel accordingly.

6) In terms of the error correction assay data presented in Figure 1E, there is something to be said about the fact that both cell lines look to have a comparable fold-increase in lagging chromosomes compared to the -ZM condition. This makes the ZM data less informative as the baseline level of lagging chromosome following STLC WO is \sim 2 fold higher in the 87y cells compared to the Ns.

We agree with the Reviewer's comment that irrespectively of using ZM treatment following STLC WO to exacerbate the frequency of segregation errors, the fold-increase in lagging chromosomes is always ~2 fold higher in 87y vs. neonatal cells. This makes the ZM data less informative but suggests that Aurora B activity does not account for the defective error correction in elderly cells, which comes in agreement with our idea that an increased number of microtubule attachments at the kinetochore of elderly cells accounts for the increase in CIN.

7) "Top" and "bottom" appear to be used incorrectly in figure legend 1G.

We have revised Figure 1G and its legend is now in accordance with what is depicted.

8) What is responsible for the increase in time from NEB-anaphase in the old cells? Does it take longer for the cells to align their chromosomes or is the duration of metaphase longer?

There are a number of mitotic defects that could be responsible for the increased mitotic duration in aged cells (*Macedo et al., Nat Commun 2018, Figure 2*). From our previous live-cell imaging efforts, we do envision that cells take more time to align their chromosomes properly, but we are not sure whether duration of metaphase is longer and how this would provide much additional insight to this study.

9) The presentation of the transcriptomics results in MCAK and Kif2b over-expressing cells was somewhat mis-leading in the text. For example, 17 (out of 20 SASP genes) were "correctly altered following over-expression of kinesin-13 proteins" gave me the impression that the transcript levels were restored to HDF N levels. Instead they moved in the right direction but were not fully "corrected". Please phrase this similarly to how it is presented for the cGAS, etc. genes where it is clearly stated that MCAK/Kif2b over-expression gave a partial rescue.

We rephrased this as per Reviewer's suggestion to clearly state that MCAK/Kif2b over-expression partially rescued SASP gene expression.

10) With regards to the adaptation to UMK57, while I agree it is not substantial, the data presented in Figure 5H shows that there is a reduction ($p < 0.05$) in centromeric pAurB levels after 96 hours in UMK57. The stats are not shown in the plot, but are the levels after 96h also reduced relative to the 24 hour condition?

Centromeric pAurB levels in elderly cells were significantly reduced ($p < 0.05$) after 96h in UMK57, even though not significantly reduced relative to 24h treatment (revised Figure 5H). Importantly our functional analysis does not indicate that this change leads to an adaptive response (Figure 5A, F).

The text in the main body of the manuscript should acknowledge this reduction as it is presently reads that "analysis.. at centromeres did not reveal substantially reduced levels in.. metaphase (Fig 5G,H) after 96 hours."

This reduction is now acknowledged in the revised version of the manuscript (lines 280-283, page 10).

Referee #3:

Logarinho and colleagues present a nice follow up to the Logarinho laboratory's previous study demonstrating that aged human fibroblasts exhibit chromosomal instability and aneuploidy, leading to senescence. This present study nicely advances these findings by demonstrating that mis-regulated mitotic proteins leading to incorrect microtubule dynamics are likely a cause of the elevated chromosome mis-segregation seen in elderly cells, since modulation of microtubule dynamics can rescue these defects.

The authors use two independent methods to correct aberrant kinetochore-microtubule attachments, taking advantage of a recently discovered small molecule potentiator of the microtubule depolymerase MCAK, and confirming these results with overexpression of GFP-tagged MCAK. Interestingly they also go on to show that the rescue of mitotic defects, aneuploidy and senescence can be long-lived (above 96 hours) in contrast to cancer cells that appear to circumvent the rescue of chromosomal instability within a short time period, as previously shown by the Compton laboratory. Overall the experimental design is well conceived, the experiments well performed and clearly presented. The manuscript is well written and clear. I recommend this for publication in EMBO reports, and have only a couple of minor points/suggestions:

We thank the Reviewer for his/her positive comments.

1. Figure 1C and 1D: what does n=50 refer to, individual inter-KT distance measurements, or cells? If individual KT measurements could they provide the number of total cells analysed too?
 N=50 refers to individual KT pairs measured in a total of 10 cells per condition. We now provide this information in Figure 1 legend.

2. Figure 2A: GFP-MCAK band in the Western blot looks very odd in the 87y fibroblasts, why is this?

We agree with the Reviewer about the quality of the GFP-MCAK band in the 87y fibroblasts and we now provide a Western blot image with increased quality in revised Figure 2A.

Could the authors provide an immunofluorescence image of the GFP-MCAK cells?

We provide now in Appendix Figure S2 fluorescence microscopy images of neonatal and octogenarian fibroblasts transduced with GFP-MCAK lentiviruses. Moreover, we provide the percentage of cells expressing GFP-MCAK following lentiviral transduction.

3. Figure 3A and 3B: It would be helpful to include an example image of the 53BP1/p21 and SA- β -Gal IF (shown nicely in the supplementary data) in the main figure.

We agree with the Reviewer's suggestion and we transferred representative images of the 53BP1/p21 and SA- β -Gal IF from the supplementary data to revised Figure 3.

2nd Editorial Decision

4 February 2020

Thank you for the submission of your revised manuscript to EMBO reports. We have now received the reports from former referee 1 and 2 that are copied below.

As you will see, both referees are very positive about the study and support publication without further revision.

Browsing through the manuscript myself, I noticed a few editorial things that we need before we can proceed with the official acceptance of your study.

REFEREE REPORTS

Referee #1:

The authors adequately responded to the suggestions/requests. This is an important study, extremely well done and I strongly support its publication.

Referee #2:

The effort the authors made to address my comments is appreciated and the resubmitted manuscript has been strengthened. I support publication of the work in EMBO Reports.

2nd Revision - authors' response

5 February 2020

The authors performed all minor editorial changes.

Corresponding Author Name: Elsa Logarinho

Manuscript Number: EMBOR-2019-49248